# BEYOND REWARD: OFFLINE PREFERENCE-GUIDED POLICY OPTIMIZATION

## ABSTRACT

In this work, we study offline preference-based reinforcement learning (PbRL), which relaxes two fundamental supervisory signals in standard reinforcement learning (online accessible transition dynamics and rewards). In other words, the agent is provided with fixed offline trajectory transitions and human preferences between pairs of trajectories. Due to the orthogonality property of rewards and dynamics, one common practice is combining prior PbRL-based reward learning objectives with off-the-shelf offline RL algorithms to bridge preference modeling and offline learning. However, such two isolated optimizations require learning a separate reward function and thus place an information bottleneck on reward learning (the bridge). As an alternative, we propose offline preference-guided policy optimization (OPPO), an end-to-end offline PbRL formulation, which jointly learns to model the preference (for finding the optimal task policy) and the offline data (for eliminating OOD). In particular, OPPO introduces an offline hindsight information matching objective and a preference modeling objective. Then, by iterating the two objectives over, we can directly extract a well-performing decision policy, avoiding a separate reward learning. We empirically show that OPPO can effectively model the offline preference and outperform prior competing baselines (including the offline RL algorithms performed over the true reward function).

## 1 INTRODUCTION

Deep reinforcement learning (RL) presents a flexible framework for learning task-oriented behaviors (Kohl & Stone, 2004; Kober & Peters, 2008; Kober et al., 2013; Silver et al., 2017; Kalashnikov et al., 2018; Vinyals et al., 2019), where the "task" is often expressed as maximizing the cumulative reward sum of trajectories generated by rollouting the learning policy in the corresponding environment. However, the above RL formulation implies two indispensable prerequisites for the training of decision policy: 1) an interactable environment and 2) a pre-specified reward function. Unfortunately, 1) online interactions with the environment can be costly and unsafe (Mihatsch & Neuneier, 2002; Hans et al., 2008; García & Fernández, 2015), and 2) designing a suitable reward function often requires expensive human effort, while the heuristic rewards often used are sometimes incapable of conveying the true intention (Hadfield-Menell et al., 2017).

To relax these requirements, previous works have either 1) focused on the offline RL formulation (the online rollout is inaccessible) (Fujimoto et al., 2019), where the learner has access to fixed offline trajectories along with a reward signal for each transition (or along with limited expert demonstrations), or 2) considered the (online) preference-based RL (PbRL) formulation (Christiano et al., 2017; Bıyık & Sadigh, 2018; Sadigh et al., 2017; Biyik et al., 2020; Lee et al., 2021a), where messages about the task objective are passed to the learner through preferences of a (human) annotator between two trajectories rather than rewards for each transition. To further progress in this setting, we propose relaxing both requirements and advocating for offline PbRL.

In the offline PbRL setting with access to an offline dataset and labeled preferences between the offline trajectories, it is straightforward to combine previous (online) PbRL methods and off-the-shelf offline RL algorithms (Shin & Brown, 2021). As shown in Fig.1 (left), we can first use the Bradley-Terry model (Bradley & Terry, 1952) to model the preference label and supervisedly learn a reward function (normally adopted in prior PbRL methods), and then train the policy with any offline RL algorithm on the transitions relabeled via the learned reward function. Intuitively, such

Figure 1: A flow diagram of previous offline PbRL algorithms (left) and our OPPO algorithm (right). Previous works require learning a separate reward function for modeling human preferences using the Bradley-Terry model. In contrast, our OPPO directly learns the policy network.

a two-step procedure allows preference modeling via a separate reward function. Fundamentally, however, learning a separate reward function that explains expert preferences does not directly instruct the policy how to act optimally. As PbRL tasks are defined by preference labels, the goal is to learn the most preferred trajectory by the annotator rather than to maximize the cumulative discounted proxy rewards of the policy rollouts. More specifically, when encountering complex tasks such as non-Markovian tasks, conveying information from preferences to the policy via the scalar rewards creates a bottleneck in policy improvement. On the other hand, if the learned reward function is miscalibrated, an isolated policy optimization may learn to exploit loopholes in the relabeled rewards, resulting in unwanted behaviors. Then why must we learn a reward function considering it possibly being not able to directly yield optimal actions?

To this end, we propose offline preference-guided policy optimization (OPPO), an end-to-end formulation that jointly models offline preferences and learns the optimal decision policy without learning a separate reward function (as shown in Fig.1 right). Specifically, we explicitly introduce a hindsight information encoder with which we further design an offline hindsight information matching objective and a preference modeling objective. Via optimizing the two objectives iteratively, we derive a contextual policy $\pi(\mathbf{a}|\mathbf{s}, \mathbf{z})$ to model the offline data and concurrently optimize an optimal contextual variable $\mathbf{z}^*$ to model the preference. In this way, the focus of OPPO is on learning a high-dimensional z-space capturing more task-related information and evaluating policies in it. Then, we arrive at the optimal policy by having the contextual policy $\pi(\mathbf{a}|\mathbf{s}, \mathbf{z}^*)$ conditioned on the learned optimal $\mathbf{z}^*$.

In summary, our contributions include 1) OPPO: a simple, stable and end-to-end offline PbRL method that avoids learning a separate reward function, 2) an instance of preference-based hindsight information matching objective and a novel preference modeling objective over the contextual, and 3) extensive experiments to show and analyze the outstanding performance of OPPO against previous competitive baselines.

## 2 RELATED WORK

**Preference-based RL.** Preference-based RL is also known as learning from human feedback. Several works have successfully utilized feedback from real humans to train RL agents (Arumugam et al., 2019; Christiano et al., 2017; Ibarz et al., 2018; Knox & Stone, 2009; Lee et al., 2021b; Warnell et al., 2017). Christiano et al. (2017) scaled preference-based reinforcement learning to utilize modern deep learning techniques, and Ibarz et al. (2018) improved the efficiency of this method by introducing additional forms of feedback such as demonstrations. Recently, Lee et al. (2021b) proposed a feedback-efficient RL algorithm by utilizing off-policy learning and pre-training. Park et al. (2022) used pseudo-labeling to utilize unlabeled segments and proposed a novel data augmentation method called temporal cropping.

**Offline RL.** To mitigate the impact of distribution shifts in offline RL, prior algorithms (a) constrain the action space (Fujimoto et al., 2019; Kumar et al., 2019a; Siegel et al., 2020), (b) incorporate value pessimism (Fujimoto et al., 2019; Kumar et al., 2020), and (c) introduce pessimism into learned dynamics models (Kidambi et al., 2020; Yu et al., 2020). Another line of work explored learning a wide behavior distribution from the offline dataset by learning a task-agnostic set

of skills, either with likelihood-based approaches (Ajay et al., 2020; Campos et al., 2020; Pertsch et al., 2020; Singh et al., 2020) or by maximizing mutual information (Eysenbach et al., 2018; Lu et al., 2020; Sharma et al., 2019). Shin & Brown (2021) tried to solve offline PbRL by simply combining previous (online) PbRL methods and off-the-shelf offline RL algorithms.

**Supervised learning in RL.** Some prior methods for reinforcement learning bear more resemblance to static supervised learning, such as Q-learning (Watkins, 1989; Mnih et al., 2013) and behavior cloning. In these cases, the resulting agent's performance is positively correlated to the quality of data used for training. On the other hand, Srivastava et al. (2019) and Kumar et al. (2019b) studied "upside-down" reinforcement learning (UDRL), seeking to model behaviors via a supervised loss conditioned on a target return. Ghosh et al. (2019) extended prior UDRL methods to perform goal reaching by taking the goal state as the condition, and Paster et al. (2020) further used an LSTM for goal-conditioned online RL settings. Chen et al. (2021) and Janner et al. (2021) solved the problem via sequence modeling. Sequence modeling enables to model behaviors without access to the reward, in a similar style to language (Radford et al., 2018) and images (Chen et al., 2020). In contrast to both supervised RL and UDRL, the purpose of our method is to search for the optimal solution supervised by a binary preference signal in the offline setting. Our method can not only work with sub-optimal demonstrations but also reveal optimal behaviors without injecting human priors about the optimal demonstration.

## 3 PRELIMINARIES

We consider reinforcement learning (RL) in a Markov decision process (MDP) described by a tuple $(\mathcal{S}, \mathcal{A}, r, P, p_0, \gamma)$, where $\mathbf{s}_t \in \mathcal{S}$, $\mathbf{a}_t \in \mathcal{A}$, and $r_t = r(\mathbf{s}_t, \mathbf{a}_t)$ denote the state, action, and reward at timestep $t$, $P(\mathbf{s}_{t+1}|\mathbf{s}_t, \mathbf{a}_t)$ denotes the transition dynamics, $p_0(\mathbf{s}_0)$ denotes the initial state distribution, and $\gamma \in [0, 1)$ denotes the discount factor. At each timestep $t$, the agent receives a state $\mathbf{s}_t$ from the environment and chooses an action $\mathbf{a}_t$ based on the policy $\pi(\mathbf{a}_t|\mathbf{s}_t)$. In the standard RL framework, the environment returns a reward $r_t$ and the agent transitions to the next state $\mathbf{s}_{t+1}$. The expected return $\mathcal{J}_r(\pi) = \mathbb{E}_{\tau \sim \pi(\tau)} \sum_{k=0}^{\infty} \gamma^k r(s_{t+k}, a_{t+k})$ is defined as the expectation of discounted cumulative rewards, where $\tau = (\mathbf{s}_0, \mathbf{a}_0, \mathbf{s}_1, \mathbf{a}_1, \dots)$, $\mathbf{s}_0 \sim p_0(\mathbf{s}_0)$, $\mathbf{a}_t \sim \pi(\mathbf{a}_t|\mathbf{s}_t)$, and $\mathbf{s}_{t+1} \sim P(\mathbf{s}_{t+1}|\mathbf{s}_t, \mathbf{a}_t)$. The agent's goal is to learn a policy $\pi$ that maximizes the expected return.

### 3.1 OFFLINE PREFERENCE-BASED REINFORCEMENT LEARNING

In this work, we assume a fully offline setting in which the agent cannot conduct online rollouts (over the MDP) during training but is provided with a static fixed dataset. The static dataset, $\mathcal{D} := \{\tau^0, \dots, \tau^N\}$, consists of some pre-collected trajectories, where each trajectory $\tau^i$ consists of a contiguous sequence of states and actions: $\tau^i := \{\mathbf{s}_0^i, \mathbf{a}_0^i, \mathbf{s}_1^i, \dots\}$. Such an offline setting is more challenging than the standard (online) setting as it removes the ability to explore the environment and collect additional feedback. Unlike imitation learning, we do not assume that the dataset comes from a single expert policy. Instead, the dataset $\mathcal{D}$ may contain trajectories collected by some suboptimal or even random behavior policies.

Further, the standard offline RL assumes the existence of reward signals for each state-action pair in $\mathcal{D}$. However, in the offline Preference-based RL (PbRL) framework, we do not assume such rewards are accessible. Instead, the agent can access offline preferences (between some pairs of trajectories $(\tau^i, \tau^j)$) that are labeled by an expert (human) annotator. Specifically, the annotator gives a feedback indicating which trajectory is preferred, i.e., $y \in \{0, 1, 0.5\}$, where 0 indicates $\tau^i \succ \tau^j$ (the event that trajectory $\tau^i$ is preferable to trajectory $\tau^j$), 1 indicates $\tau^j \succ \tau^i$ ($\tau^j$ is preferable to $\tau^i$), and 0.5 implies an equally preferable case. Each feedback is stored in a labeled offline dataset $\mathcal{D}_\succ$ as a triple $(\tau^i, \tau^j, y)$. Given these preferences, the goal of PbRL is to find a policy $\pi(\mathbf{a}_t|\mathbf{s}_t)$ that maximizes the expected return $\mathcal{J}_{r_\psi}$, under the hypothetical reward function $r_\psi(\mathbf{s}_t, \mathbf{a}_t)$ consistent with human preferences. To enable this, previous works learn a reward function $r_\psi(\mathbf{s}_t, \mathbf{a}_t)$ and use the Bradley-Terry model (Bradley & Terry, 1952) to model the human preference, expressed here as a logistic function:

$$P[\tau^i \succ \tau^j] = \text{logistic}(\sum_t r_\psi(\mathbf{s}_t^i, \mathbf{a}_t^i) - \sum_t r_\psi(\mathbf{s}_t^j, \mathbf{a}_t^j)), \tag{1}$$

where $(\mathbf{s}_t^i, \mathbf{a}_t^i) \sim \tau^i$, $(\mathbf{s}_t^j, \mathbf{a}_t^j) \sim \tau^j$. Intuitively, this can be interpreted as the assumption that the probability of preferring a trajectory depends exponentially on the cumulative reward over the trajectory labeled by an underlying reward function. The reward function is then updated by minimizing the following cross-entropy loss:

$$-\mathbb{E}_{(\tau^i, \tau^j, y) \sim \mathcal{D}_{\succ}} \left[ (1 - y) \log P[\tau^i \succ \tau^j] + y \log P[\tau^j \succ \tau^i] \right]. \tag{2}$$

With the learned reward function $r_\psi$ used to label each transition in the dataset, we can adopt an off-the-shelf offline RL algorithm to enable policy learning.

## 3.2  Hindsight Information Matching

Beyond the typical iterative (offline) RL framework, information matching (IM) (Furuta et al., 2021) has been recently studied as an alternative problem specification in (offline) RL. The objective of IM in RL is to learn a contextual policy $\pi(\mathbf{a}|\mathbf{s}, \mathbf{z})$ whose trajectory rollouts satisfy some (pre-defined) desired information statistics value $\mathbf{z}$:

$$\min_\pi \mathbb{E}_{\mathbf{z} \sim p(\mathbf{z}), \tau_{\mathbf{z}} \sim \pi(\tau_{\mathbf{z}})} \left[ \ell \left( \mathbf{z}, I(\tau_{\mathbf{z}}) \right) \right], \tag{3}$$

where $p(\mathbf{z})$ is a prior, $\pi(\tau_{\mathbf{z}})$ denotes the trajectory distribution generated by rolling out $\pi(\mathbf{a}|\mathbf{s}, \mathbf{z})$ in the environment. $I(\tau)$ is a function capturing the statistical information of a trajectory $\tau$, such as the distribution statistics of state and reward, like mean, variance (Wainwright et al., 2008), and $\ell$ is a loss function.

On one hand, if we set $p(\mathbf{z})$ as a prior distribution, optimizing Eq.3 corresponds to performing unsupervised (online) RL to learn a set of skills (Eysenbach et al., 2018; Sharma et al., 2019). On the other hand, if we set $p(\mathbf{z})$ as statistical information of a given off-policy trajectory (or state-action) distribution $\mathcal{D}(\tau)$ (or $\mathcal{D}(\mathbf{s}, \mathbf{a})$), Eq.3 corresponds to an objective for hindsight information matching in (offline) RL. For example, HER (Andrychowicz et al., 2017) and return-conditioned RL (upside-down RL (Srivastava et al., 2019; Kumar et al., 2019b; Chen et al., 2021; Janner et al., 2021)) use the above concept of hindsight: specifying any trajectory $\tau$ in the dataset as the hindsight target and setting the information $\mathbf{z}$ in Eq.3 as $I(\tau)$. Then, we provide the $I(\cdot)$-driven hindsight information matching (HIM) objective:

$$\min_\pi \mathbb{E}_{\tau \sim \mathcal{D}(\tau), \tau_{\mathbf{z}} \sim \pi(\tau_{\mathbf{z}})} \left[ \ell \left( I(\tau), I(\tau_{\mathbf{z}}) \right) \right], \tag{4}$$

where $\mathbf{z} := I(\tau)$. In HER, we can set $I(\tau)$ as the final state in trajectory $\tau$; in reward-conditional RL, we set $I(\tau)$ as the return of trajectory $\tau$. Thus, we can use the hindsight information $\mathbf{z} := I(\tau)$ to provide supervision for training the contextual policy $\pi(\mathbf{a}|\mathbf{s}, \mathbf{z})$. However, in offline settings, sampling $\tau_{\mathbf{z}}$ from $\pi(\tau_{\mathbf{z}})$ is not accessible. Thus, we are required to model the environment transition dynamics besides $I(\cdot)$-driven hindsight information modeling. That is to say, we need to model the trajectory itself, $i.e.$, $\min_\pi \mathbb{E}_{\tau \sim \mathcal{D}(\tau), \tau_{\mathbf{z}} \sim \pi(\tau_{\mathbf{z}})} \left[ \ell \left( \tau, \tau_{\mathbf{z}} \right) \right]$. Then, we provide the overall offline HIM objective:

$$\min_\pi \mathbb{E}_{\tau \sim \mathcal{D}(\tau), \tau_{\mathbf{z}} \sim \pi(\tau_{\mathbf{z}})} \left[ \ell \left( I(\tau), I(\tau_{\mathbf{z}}) \right) + \ell \left( \tau, \tau_{\mathbf{z}} \right) \right]. \tag{5}$$

To give an intuitive understanding of the above objective, here we provide a simple example: considering hindsight $I(\cdot)$ being the return of a trajectory, optimizing $\ell \left( I(\tau), I(\tau_{\mathbf{z}}) \right)$ ensures the generated $\tau_{\mathbf{z}}$ will reach the same return as $\tau = I^{-1}(\mathbf{z})$. However, in the offline setting, we must ensure the generated $\tau_{\mathbf{z}}$ stay in support of the offline data, eliminating the out-of-distribution (OOD) issues. Thus we minimize $\ell \left( \tau, \tau_{\mathbf{z}} \right)$ approximately. In implementation, directly optimizing $\ell \left( \tau, \tau_{\mathbf{z}} \right)$ is enough to ensure the hindsight information is matched, $e.g.$, $\ell \left( I(\tau), I(\tau_{\mathbf{z}}) \right) < \epsilon$. Here, we explicitly formalizes the $\ell \left( I(\tau), I(\tau_{\mathbf{z}}) \right)$ term with particular emphasis on the requisite of hindsight information matching objective and meanwhile highlight the difference, see Section 4, between the above HIM objective (taking $I(\cdot)$ as a prior) and our OPPO formulation (requiring learning $I_\theta(\cdot)$).

By optimizing Eq.5, we can obtain a contextual policy $\pi(\mathbf{a}|\mathbf{s}, \mathbf{z})$. In the evaluation phase, the optimal policy $\pi(\mathbf{a}|\mathbf{s}, \mathbf{z}^*)$ can be specified by conditioning the policy on a selected target $\mathbf{z}^*$. For example, Decision Transformer (Chen et al., 2021) takes the desired performance as the target $\mathbf{z}^*$(e.g., specify maximum possible return to generate expert behavior); RvS-G (Emmons et al., 2021) takes the goal state as the target $\mathbf{z}^*$.

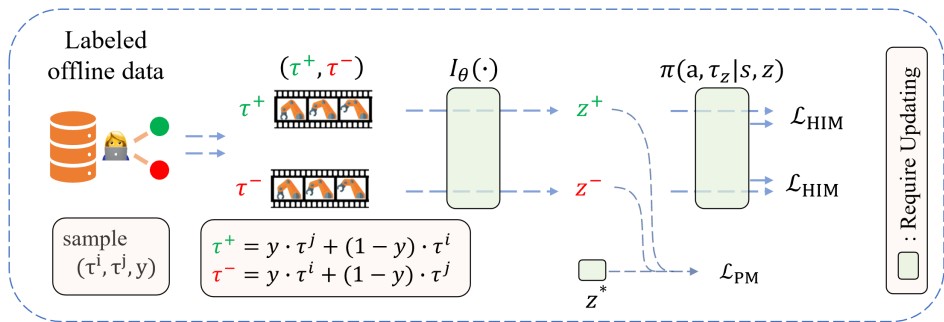

Figure 2: OPPO first maps offline trajectories (both positive $\tau^+$ and negative $\tau^-$) to a latent space via the hindsight information extractor $I_\theta$. It then optimizes the offline HIM objective $\mathcal{L}_{\text{HIM}}$. Finally, the belief of the optimal hindsight information $\mathbf{z}^*$ is updated to model the human preference with objective $\mathcal{L}_{\text{PM}}$. Meanwhile, the preference modeling loss also regularizes the learning of the hindsight information extractor $I_\theta$.

## 4 OPPO: OFFLINE PREFERENCE-GUIDED POLICY OPTIMIZATION

In this section, we present our method, OPPO (offline preference-guided policy optimization), that adopts the hindsight information matching (HIM) objective in Section 3.2 to model an offline contextual policy $\pi(\mathbf{a}|\mathbf{s}, \mathbf{z})$, and introduces a triplet loss to model the human preference as well as the optimal contextual variable $\mathbf{z}^*$. At testing, we condition the policy on the optimal $\mathbf{z}^*$ and thus conduct rollout with $\pi(\mathbf{a}|\mathbf{s}, \mathbf{z}^*)$. In principle, OPPO can be paired with any PbRL settings, including both online and offline. In the scope of our analysis and experiments, however, we focus on the offline setting to decouple exploration difficulties in online RL.

### 4.1 HIM-DRIVEN POLICY OPTIMIZATION

As described in Section 3.1, to directly implement the off-the-shelf offline RL algorithms, previous works in PbRL explicitly learn a reward function with Eq.2 (as shown in Fig.1 left). As an alternative to such a two-step approach, we seek to learn the policy directly from the preference-labeled offline dataset (as shown in Fig.1 right). Inspired by the offline HIM objective in Section 3.2, we propose to learn a contextual policy $\pi(\mathbf{a}|\mathbf{s}, \mathbf{z})$ in the offline PbRL setting. Assuming $I_\theta$ is a (learnable) network that encodes the hindsight information in PbRL, we formulate the following objective:

$$\min_{\pi, I_\theta} \mathcal{L}_{\text{HIM}} := \mathbb{E}_{\tau \sim \mathcal{D}(\tau), \tau_{\mathbf{z}} \sim \pi(\tau_{\mathbf{z}})} \left[ \ell \left( I_\theta(\tau), I_\theta(\tau_{\mathbf{z}}) \right) + \ell \left( \tau, \tau_{\mathbf{z}} \right) \right], \quad (6)$$

where $\mathbf{z} := I_\theta(\tau)$. Note that Eq.6 is a different instantiation of Eq.5 where we learn the hindsight information extractor $I_\theta(\cdot)$ in the PRBL setting, while previous (offline) RL algorithms normally set $I(\cdot)$ to be a prior (Chen et al., 2021; Emmons et al., 2021). Such an encoder-decoder structure is now similar with Bi-directional Decision Transformer (BDT) proposed by (Furuta et al., 2021) for offline imitation learning. However, since expert demonstrations are missing in the PbRL setting, in Section 4.2, we propose to use the preference labels in $\mathcal{D}_\succ$ to extract hindsight information.

### 4.2 PREFERENCE MODELING

To make the hindsight information $I_\theta(\tau)$ in Eq.6 to match the preference information in the (labeled) dataset $\mathcal{D}_\succ$, we construct the following preference modeling objective inspired by the contrastive loss in metric learning:

$$\min_{\mathbf{z}^*, I_\theta} \mathbb{E}_{(\tau^i, \tau^j, y) \sim \mathcal{D}_\succ} \left[ \ell(\mathbf{z}^*, \mathbf{z}^+) - \ell(\mathbf{z}^*, \mathbf{z}^-) \right], \quad (7)$$

where $\mathbf{z}^+$ and $\mathbf{z}^-$ represent the embedding of the preferable (positive) trajectory $I_\theta(y\tau^j + (1-y)\tau^i)$ and that of the less preferable (negative) trajectory $I_\theta(y\tau^i + (1-y)\tau^j)$, respectively. Closing to the idea of using regret for modeling preference (Knox et al., 2022; Chen et al., 2022), our basic assumption of designing the objective in Eq.7 is that humans normally conduct two-level comparisons

---

**Algorithm 1** OPPO: Offline Preference-guided Policy Optimization

---

**Require:** Dataset $\mathcal{D} := \{\tau\}$ and labeled dataset $\mathcal{D}_{\succ} := \{(\tau^i, \tau^j, y)\}$, where $\tau^i \in \mathcal{D}$ and $\tau^j \in \mathcal{D}$.
**Return:** $\pi(\mathbf{a}|\mathbf{s}, \mathbf{z})$ and $\mathbf{z}^*$.

1: Initialize policy network $\pi(\mathbf{a}|\mathbf{s}, \mathbf{z})$, hindsight information extractor $I_\theta : \tau \to \mathbf{z}$, and the optimal contextual embedding $\mathbf{z}^*$.
2: **while** not converged **do**
3:      Sample a batch of trajectories from $\mathcal{D}$: $\{\tau\}_\text{B} \sim \mathcal{D}$.
4:      Update $\pi(\mathbf{a}|\mathbf{s}, \mathbf{z})$ and $I_\theta(\cdot)$ with sampled $\{\tau\}_\text{B}$ using $\mathcal{L}_\text{HIM}$.
5:      Sample a batch of preferences from $\mathcal{D}_{\succ}$: $\{(\tau^i, \tau^j, y)\}_\text{B} \sim \mathcal{D}_{\succ}$.
6:      Update $I_\theta(\cdot)$ and the optimal $\mathbf{z}^*$ with sampled $\{(\tau^i, \tau^j, y)\}_\text{B}$ using $\mathcal{L}_\text{PM}$.
7: **end while**

---

before giving preferences between two trajectories $(\tau^i, \tau^j)$: 1) separately judging the similarity between trajectory $\tau^i$ and the hypothetical optimal trajectory $\tau^*$, i.e. $-\ell(\mathbf{z}^*, \mathbf{z}^i)$, and the similarity between trajectory $\tau^j$ and the hypothetical optimal one $\tau^*$, $-\ell(\mathbf{z}^*, \mathbf{z}^j)$, and 2) judging the difference of the above two similarities ($-\ell(\mathbf{z}^*, \mathbf{z}^i)$ vs. $-\ell(\mathbf{z}^*, \mathbf{z}^j)$) and setting the trajectory with the higher similarity as the preferred one. Hence, optimizing Eq.7 guarantees finding the optimal embedding that is more similar to $\mathbf{z}^+$ and less similar to $\mathbf{z}^-$. To clarify, $\mathbf{z}^*$ is the catresponding contextual information for $\tau^*$, whereas $\tau^*$ will always be preferred comparing to any offline trajectories in the datasets.

In practice, to robustify the preference modeling, we optimize the following objective using the triplet loss in place of the objective in Eq.7:

$$\min_{\mathbf{z}^*, I_\theta} \mathcal{L}_\text{PM} := \mathbb{E}_{(\tau^i, \tau^j, y) \sim \mathcal{D}_{\succ}} \Big[ \max(\ell(\mathbf{z}^*, \mathbf{z}^+) - \ell(\mathbf{z}^*, \mathbf{z}^-) + \text{margin}, 0) \Big]. \tag{8}$$

It is worth mentioning that the posterior of the optimal embedding $\mathbf{z}^*$ and the hindsight information extractor $I_\theta(\cdot)$ are updated alternatively to ensure learning stability. A better estimate of the optimal embedding helps the encoder to extract features to which the human labeler pay more attention, while a better hindsight information encoder, on the other hand, accelerates the search process for the optimal trajectory in the high-level embedding space. In this way, the loss function for the encoder consists of two parts: 1) a hindsight information matching loss in a supervised style as in Eq.6 and 2) a triplet loss as in Eq.8 to better incorporate the binary supervision provided by the preference-labeled dataset.

**In summary**, OPPO learns a contextual policy $\pi(\mathbf{a}|\mathbf{s}, \mathbf{z})$, a context (hindsight information) encoder $I_\theta(\tau)$, and the optimal context variable, $\mathbf{z}^*$, for the optimal trajectory $\tau^*$. Algorithm 1 details the training of OPPO. The entire process is summarized as follows: 1) we sample a batch of trajectories from the dataset $\mathcal{D}$, 2) in Line 4, use Eq.6 (the hindsight information matching loss) to update $\pi(\mathbf{a}|\mathbf{s}, \mathbf{z})$ and $I_\theta(\cdot)$ based on sampled trajectories. Consequently, given the $\mathbf{z}$ extracted out of an offline trajectory by the extractor, the policy is able to reconstruct it. 3) Then sample a batch of preferences from the labeled dataset $\mathcal{D}_{\succ}$, and finally, 4) in Line 6, update $I_\theta(\cdot)$ and $\mathbf{z}^*$ based on the sampled $\{(\tau^i, \tau^j, y)\}_\text{B}$ using Eq.8, making the optimal embedding $\mathbf{z}^*$ near to the more preferred trajectory $\mathbf{z}^+$, and meanwhile further away from the less preferred trajectory $\mathbf{z}^-$.

Compared with previous PbRL works (first learning a reward function with Eq.2 and then learning offline policy with off-the-shelf offline RL algorithms), OPPO learns the optimal (offline) policy ($\pi(\mathbf{a}|\mathbf{s}, \mathbf{z}*)$) directly and thus avoids the potential information bottleneck caused by the limited information capacity of scalar reward assignment. Compared with the HIM-based offline RL algorithms (*e.g.*, Decision Transformer (Chen et al., 2021) and RvS-G (Emmons et al., 2021)), at the testing phase, OPPO does not need to manually specify the target contextual variable for the rollout policy $\pi(\mathbf{a}|\mathbf{s}, \cdot)$.

## 5 EXPERIMENTS

In this section, we evaluate and compare OPPO to other baselines in the offline PbRL setting. A central premise behind the design of OPPO is that the learned hindsight information encoder $I_\theta(\cdot)$ can capture preferences over different trajectories, as described by Eq.8. Our experiments are therefore designed to answer the following questions:

1) Does OPPO truly capture these types of preferences? In other words, does the learned $\mathbf{z}$-space (encoded by the learned $I_\theta(\cdot)$) align with given preferences?

2) Can the learned optimal contextual policy $\pi(\mathbf{a}|\mathbf{s}, \mathbf{z}^*)$ obtain a better performance than $\pi(\mathbf{a}|\mathbf{s}, \mathbf{z})$ conditioned on all the other contextual variables $\mathbf{z} \in \{I_\theta(\tau)|\tau \in \mathcal{D}\}$?

3) Can OPPO also achieve competitive performance against other offline (PbRL) algorithms?

4) What benefits, if any, do we gain from designing the end-to-end offline preference learning, *i.e.*, iteratively conducting offline data modeling (Eq.6) and preference modeling (Eq.8)?

To answer the above questions, we evaluate OPPO on the continuous control tasks from the D4RL benchmark (Fu et al., 2020). Specifically, we choose Hopper, Walker and Halfcheetah as three base environments, with medium, medium-replay, medium-expert as dataset for each environment. To build our labeled offline preference dataset $\mathcal{D}_\succ := \{(\tau^i, \tau^j, y)\}$, following prior PbRL benchmark (Lee et al., 2021a), we use the following stochastic preference model (to simulate a human annotator and) to label preference between trajectories $(\tau^i, \tau^j)$:

$$y \sim P[\tau^i \succ \tau^j; \beta, \gamma] = \text{logistic}(\beta \sum_{t=1}^{H} \gamma^{H-t} r(\mathbf{s}_t^i, \mathbf{a}_t^i) - \beta \sum_{t=1}^{H} \gamma^{H-t} r(\mathbf{s}_t^j, \mathbf{a}_t^j)), \qquad (9)$$

where factors $\gamma$ and $\beta$ are used to model myopic and rational behaviors (of human annotator) respectively and are both set to 1 in our experiment.

## 5.1 DOES THE LEARNED $\mathbf{z}$-SPACE ALIGN WITH GIVEN PREFERENCES?

In this subsection, we probe that OPPO can enable well-aligned preferences over the $\mathbf{z}$-space encoded by the learned $I_\theta$. We first sample random trajectories from the offline dataset $\mathcal{D}$, and encode them with the learned $I_\theta$, and utilize t-SNE (van der Maaten & Hinton, 2008) as a tool to visualize the encoded $\mathbf{z}$, shown in Fig.3. The learned optimal $\mathbf{z}^*$ is marked with an orange dot. Besides, we also mark the (embedding of) optimal trajectory in D4RL expert dataset, generated by the learned online optimal policy, with a red dot ($\mathbf{z}^{**}$).

According to Eq.8, embeddings near the actual optimal $\mathbf{z}^{**}$ in $\mathbf{z}$-space means they are more preferable implied by the preference label. Comparing the sampled trajectories (embeddings), we find OPPO successfully captures the preference. As illustrated in Fig.3, the trajectories (embeddings) that are near $\mathbf{z}^{**}$ often have high returns (points with a deeper color). Further, we observe that our learned optimal $\mathbf{z}^*$ constantly stays close to actual optimal $\mathbf{z}^{**}$, which suggests that our learned $\mathbf{z}^*$ preserves near-optimal behaviors. Thus, it gives justification that OPPO can make meaningful preference modeling.

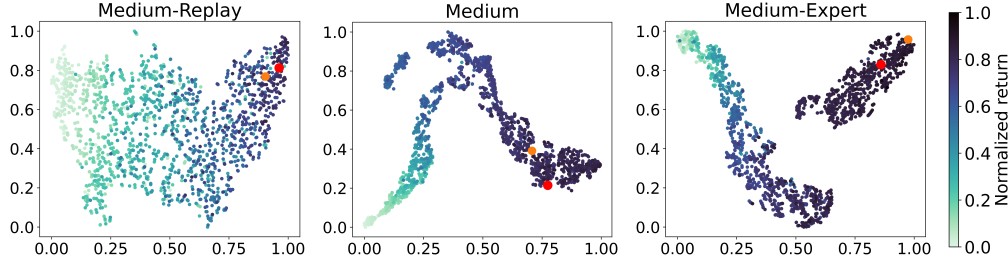

Figure 3: We utilize t-SNE to visualize the $\mathbf{z}$-space learned in Hopper environment, encoded with a well-trained $I_\theta(\cdot)$, including the embedding of random trajectories in $\mathcal{D}$, our learned $\mathbf{z}^*$ ("orange dot") and the actual optimal $\mathbf{z}^{**}$ ("red dot"), embedding of the best trajectory/policy learned with online reinforcement learning method. Color of the points represent the normalized return of the corresponding trajectory $\tau$.

## 5.2 CAN $\pi(\mathbf{a}|\mathbf{s}, \mathbf{z})$ CONDITIONED THE LEARNED $\mathbf{z}^*$ ENABLE BETTER PERFORMANCE?

Fig.3 shows that our learned $I_\theta(\cdot)$ can produce a well-aligned embedding $\mathbf{z}$-space exhibiting effective preference modeling across (embeddings of) trajectories. More importantly, embeddings'

Table 1: Comparison of (normalized) performance when rollouting the contextual policy $\pi(\mathbf{a}|\mathbf{s}, \cdot)$ conditioned on different embeddings ($\mathbf{z}^*$, $\mathbf{z}_{\text{high}}$, and $\mathbf{z}_{\text{low}}$).

| Environment | Dataset | $\mathbf{z}^*$ | $\mathbf{z}_{\text{high}}$ | $\mathbf{z}_{\text{low}}$ |
|---|---|---|---|---|
| Hopper | Medium-Expert | **108.0 $\pm$ 5.1** | $94.2 \pm 24.3$ | $79.1 \pm 28.8$ |
| | Medium | **86.3 $\pm$ 3.2** | $55.8 \pm 7.9$ | $51.6 \pm 13.8$ |
| | Medium-Replay | **88.9 $\pm$ 2.3** | $78.6 \pm 26.3$ | $26.6 \pm 15.2$ |
| Walker | Medium-Expert | $105.0 \pm 2.4$ | **106.5 $\pm$ 9.1** | $93.4 \pm 7.4$ |
| | Medium | **85.0 $\pm$ 2.9** | $64.9 \pm 24.9$ | $72.6 \pm 10.6$ |
| | Medium-Replay | **71.7 $\pm$ 4.4** | $55.7 \pm 24.8$ | $6.8 \pm 1.7$ |
| Halfcheetah | Medium-Expert | **89.6 $\pm$ 0.8** | $48.3 \pm 14.4$ | $42.6 \pm 2.6$ |
| | Medium | **43.4 $\pm$ 0.2** | $42.5 \pm 3.9$ | $42.4 \pm 3.2$ |
| | Medium-Replay | **39.8 $\pm$ 0.2** | $35.6 \pm 8.5$ | $33.9 \pm 9.2$ |
| **Sum** | | 717.7 | 581.9 | 448.9 |

preference property should be preserved when we condition the embedding on the learned contextual policy $\pi(\mathbf{a}|\mathbf{s}, \cdot)$. In other words, $I_\theta(\cdot)$ transfers the preference relationship from $(\tau^i, \tau^j)$ to $(\ell(\mathbf{z}^i, \mathbf{z}^*), \ell(\mathbf{z}^j, \mathbf{z}^*))$; further, rollouting the contextual policy $\pi(\mathbf{a}|\mathbf{s}, \cdot)$, $(\tau_{\mathbf{z}^i}, \tau_{\mathbf{z}^j})$ should similar preserve the above preference relationship.

To show that, we compare the performance of rollouts by the contextual policy $\pi(\mathbf{a}|\mathbf{s}, \cdot)$ conditioned on different embeddings. In Table 1, we choose three contextual embeddings: $\mathbf{z}^*$, $\mathbf{z}_{\text{high}}$ (embedding of the trajectory with the highest return in $\mathcal{D}$), and $\mathbf{z}_{\text{low}}$ (embedding of the trajectory with the lowest return in $\mathcal{D}$) and provide respective rollout performances (averaged over 3 seeds). We discover that the contextual $\pi(\mathbf{a}|\mathbf{s}, \mathbf{z})$ conditioned on $\mathbf{z}$ with a high (or low) return (of corresponding trajectory $\tau = I_\theta^{-1}(\mathbf{z})$) obtains an actual high (or low) return when rollouting this policy in the environment, *e.g.*, $\pi(\mathbf{a}|\mathbf{s}, \mathbf{z}_{\text{high}})$ performs better than $\pi(\mathbf{a}|\mathbf{s}, \mathbf{z}_{\text{low}})$ (thus preserving the hindsight preference relationship). Further, when conditioned on the learned optimal $\mathbf{z}^*$, $\pi(\mathbf{a}|\mathbf{s}, \mathbf{z}^*)$ produces the best performance over that conditioned on all other offline embeddings. Notice that our learned optimal $\pi(\mathbf{a}|\mathbf{s}, \mathbf{z}^*)$ performs better than the contextual policy $\pi(\mathbf{a}|\mathbf{s}, \mathbf{z}_{\text{high}})$. This result implies that the trajectory of our optimal policy is better than any trajectories in the offline dataset.

## 5.3 CAN OPPO ACHIEVE COMPETITIVE PERFORMANCE ON OFFLINE (PBRL) BENCHMARK?

We have shown that OPPO produces a near-optimal embedding $\mathbf{z}^*$, and the learned contextual policy $\pi(\mathbf{a}|\mathbf{s}, \cdot)$ can preserve the hindsight preference. This subsection investigates whether the optimal policy $\pi(\mathbf{a}|\mathbf{s}, \mathbf{z}^*)$ can achieve competitive performance on the offline (PBRL) benchmark. For comparison, we introduce three offline (PbRL) baselines: 1) DT+$r$: performing Decision Transformer with ground-truth reward function, and the results are run by us; 2) DT+$r_\psi$: performing Decision Transformer with a learned reward function (using Eq.2); 3) CQL+$r$: performing CQL with ground-truth reward function, reported from the original paper; 4) BC: performing bahavior cloning on the dataset, the results are reported from (Chen et al., 2021);

Table 2: Performance comparison between OPPO and 3 offline (PbRL) baselines (DT+$r$, DT+$r_\psi$, and CQL+$r$) in D4RL Gym-Mujoco tasks, where results are reported over 3 seeds.

| Environment | Dataset | Ours | DT+$r$ | DT+$r_\psi$ | CQL+$r$ | BC |
|---|---|---|---|---|---|---|
| Hopper | Medium-Expert | **108.0 $\pm$ 5.1** | **111.0 $\pm$ 0.5** | $95.6 \pm 27.3$ | **111.0** | 79.6 |
| | Medium | **86.3 $\pm$ 3.2** | $76.6 \pm 3.9$ | $73.3 \pm 3.0$ | 58.0 | 63.9 |
| | Medium-Replay | **88.9 $\pm$ 2.3** | **87.8 $\pm$ 4.7** | $72.5 \pm 22.2$ | 48.6 | 27.6 |
| Walker | Medium-Expert | $105.0 \pm 2.4$ | **109.2 $\pm$ 0.3** | **109.7 $\pm$ 0.1** | 98.7 | 36.6 |
| | Medium | **85.0 $\pm$ 2.9** | $80.9 \pm 3.1$ | $81.1 \pm 2.1$ | 79.2 | 77.3 |
| | Medium-Replay | $71.7 \pm 4.4$ | **79.6 $\pm$ 3.1** | **80.4 $\pm$ 4.4** | 26.7 | 36.9 |
| HalfCheetah | Medium-Expert | **89.6 $\pm$ 0.8** | $86.8 \pm 1.3$ | **88.4 $\pm$ 0.7** | 62.4 | 59.9 |
| | Medium | **43.4 $\pm$ 0.2** | $43.4 \pm 0.1$ | $43.2 \pm 0.2$ | **44.4** | 43.1 |
| | Medium-Replay | $39.8 \pm 0.2$ | $39.2 \pm 0.3$ | $38.8 \pm 0.3$ | **46.2** | 4.3 |
| **Sum** | | **717.7** | 714.5 | 683.0 | 575.2 | 429.2 |

In Table 2, we provide the comparison results. 1) OPPO has retained a comparable performance against the Decision Transformer trained using true rewards. OPPO is a PbRL approach requiring only (human) preferences, which have a more flexible and straightforward form of supervision in the real world. 2) Although DT+$r_\psi$ also shows competitive results in these benchmarks, such a method needs a target of return-to-go determined by the human prior. [1]. Our method, in contrast, avoids the need of such a prior target by searching across the z-space. We argue that this searching method brings advantages as rewards are usually hard to obtain in real-world RL applications. And preferences are the only information easily accessible for training and **deploying** an RL method.

### 5.4 Do we get any benefits by iterating offline HIM and preference modeling?

Here, we conduct an ablation study to analyze the benefit of iterating $\mathcal{L}_{\text{HIM}}$ and $\mathcal{L}_{\text{PM}}$ (for updating $I_\theta$). Firstly, we removed $I_\theta$ from $\partial\mathcal{L}_{\text{PM}}/\partial\theta$ and only left the optimal embedding $\mathbf{z}^*$ to be updated in Eq.8. Then, we continue to visualize the embedding z-space for this ablation setting (OPPO-a), and we show the t-SNE visualization in Fig.4. By comparing Fig.4 to Fig.3, we can see that the preference relationship in the embedding space (learned with

Table 3: Ablation study

| Task | OPPO | OPPO-a |
|---|---|---|
| Hopper | **88.9 ± 2.3** | 78.3 ± 7.1 |
| Walker | **71.7 ± 4.4** | 66.3 ± 1.6 |
| HalfCheetah | **39.8 ± 0.2** | 39.6 ± 0.1 |
| **Sum** | **200.4** | 184.2 |

OPPO-a) is all shuffled. In a less expressive **z**-space, it is challenging to model the preference and find the optimal $\mathbf{z}^*$. Further, as shown in Table 3, the comparison results of medium-replay tasks confirm that such an ablation does cause a performance degradation.

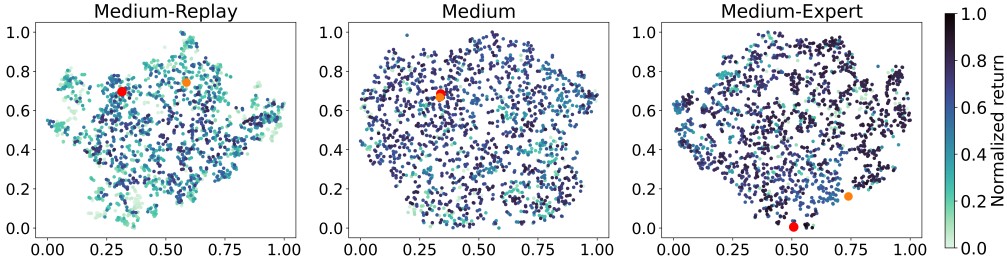

Figure 4: t-SNE visualization of the embedding space learned with OPPO-a in Hopper environment.

## 6 Conclusion

This paper introduces offline preference-guided policy optimization (OPPO), an end-to-end offline PbRL method. Unlike the previous PbRL approaches that learn policy from a pseudo reward function (thus, learning a separate reward function is a prerequisite), OPPO directly optimizes the policy in a high-level embedding space. To enable that, we suggest learning a hindsight information encoder network and using it to design an offline hindsight information matching (HIM) objective and a preference modeling objective. Empirically, we show iterating the above two objectives can produce meaningful and preference-aligned embeddings. Moreover, conditioned on the learned optimal embedding, our HIM-based contextual policy can achieve competitive performance on standard offline (PbRL) tasks.

Through the visualization, we demonstrate that the **z**-space learned by the encoder is informative and visually interpretable. Besides, the ablation study proves that a preference-guided embedding space could improve task performance asymptotically by a non-neglectable margin. Moreover, OPPO can find a contextual variable to represent the embedding of the optimal trajectory, where the resulting trajectory is better than any offline trajectory in the dataset. Last but not least, in the offline setting with environment interaction disabled, our method can acquire the optimal behaviors using binary preference labels between sub-optimal trajectories. As shown in the experiment results, OPPO achieves a competitive performance over DT trained using either true rewards or pseudo rewards.

---

[1]Preference-based relabelled rewards only participate in training phase. During the evaluation phase of DT+$r_\psi$, we pass in the same target return-to-go value as in the original DT paper.

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

# A   APPENDIX

## A.1   IMPLEMENTATION DETAILS

**Codebase.**   Our code is based on Decision Transformer: `https://github.com/kzl/decision-transformer`. We provide our source code in the supplementary material.

**OpenAI Gym.**   We choose the OpenAI Gym continuous control tasks from the D4RL benchmark (Fu et al., 2020). The different dataset settings are described below.

- Medium: 1 million timesteps generated by a "medium" policy that achieves approximately one-third the score of an expert policy.
- Medium-Replay: the replay buffer of an agent trained to the performance of a medium policy (approximately 25k-400k timesteps in our environments).
- Medium-Expert: 1 million timesteps generated by the medium policy concatenated with 1 million timesteps generated by an expert policy.

For details of these environments and datasets, please refer to D4RL for more information.

**Training Objectives**   In our experiment, we consolidate $\ell$ in Eq.6 as MSE Loss and in Eq.8 as Euclidean Distance. In this case, we model $\mathbf{z}^*$ as a point in the $\mathbf{z}-$space and the similarity measure $\ell$ is $L2$ distance. Besides, there is an alternative option to model $\mathbf{z}^*$ as a point sampled from a learned distribution in the $\mathbf{z}-$space, where $\ell$ is a measurement between two distributions such as the KL divergence.

Also, we add a normalization loss $\mathcal{L}_{\text{norm}}$ to constraint the L2 norm of all embeddings generated by hindsight information extractor $I_\theta$.

$$\mathcal{L}_{\text{total}} := \mathcal{L}_{\text{HIM}} + \alpha\mathcal{L}_{\text{PM}} + \beta\mathcal{L}_{\text{norm}} \tag{10}$$

During Offline HIM phase, we weighted sum all these 3 losses as in Eq.10 (with ratios listed in Table 4) and perform backpropagation, while in Preference Modeling phase, only $\mathcal{L}_{\text{PM}}$ is computed and backpropagated.

| Hyperparameter | Value |
|---|---|
| $\alpha$ | 0.25 for halfcheetah-medium-expert |
| | 0.5 for others |
| $\beta$ | 0.05 for halfcheetah-medium-expert |
| | 0.1 for others |

Table 4: Hyperparameters of coefficients of combined losses during Offline HIM.

**Architecture & Implementation Details**   The architecture overview of OPPO is shown in Fig.2. OPPO models the hindsight information extractor $I_\theta$ as an encoder network $I_\theta : \tau \rightarrow \mathbf{z}$, we use the BERT architecture. And similar to Chen et al. (2021), we use the GPT architecture to model $\pi(\mathbf{a}|\mathbf{s}, \mathbf{z})$, which predicts future actions via autoregressive modeling.

**Hyperparameters**   Our hyperparameters on all tasks are shown below in Table 5 and Table 6. Models were trained for $10^5$ gradient steps using the AdamW optimizer Loshchilov & Hutter (2017) following PyTorch defaults.

**Computational resources.**   The experiments were run on a computational cluster with 20x GeForce RTX 2080 Ti, and 4x NVIDIA Tesla V100 32GB for 20 days.

## A.2   ADDITIONAL RESULTS

**More visualization results on z-space.**   We further show the t-sne results of OPPO and the corresponding ablation study in the setting described in Section 5.1 and Section 5.4 in walker and

| Hyperparameter | Value |
|---|---|
| Number of dimensions | 8 for halfcheetah |
| | 16 for others |
| Amount of feedback | 50k |
| Type of optimizer | AdamW |
| Learning rate | $10^{-2}$ for halfcheetah-medium-expert |
| | $10^{-3}$ for others |
| Weight decay | $10^{-4}$ |
| Margin | 1 |

Table 5: Hyperparameters of $\mathbf{z}^*$ searching for OpenAI Gym experiments.

| Hyperparameter | Value |
|---|---|
| Number of layers | 3 |
| Number of attention heads | 2 for encoder transformer |
| | 1 for decision transformer |
| Embedding dimension | 128 |
| Nonlinearity function | ReLU |
| Batch size | 64 |
| context length K | 20 |
| Dropout | 0.1 |
| Learning rate | $10^{-4}$ |
| Grad norm clip | 0.25 |
| Weight decay | $10^{-4}$ |
| Learning rate decay | Linear warmup for first $10^5$ training steps |

Table 6: Hyperparameters of Transformer for OpenAI Gym experiments.

halfcheetah environments. By comparing Fig.6 to Fig.5, we discover that the structure of $\mathbf{z}$-space significantly collapses in eight out of nine environments (except for halfcheetah medium-replay). More specifically, we can no longer recognize the distribution pattern and clusters emerged in Fig.5, while such an observation is in line with our conclusion in the main text.

However, it is also worth noting that the performance of OPPO-a in D4RL benchmark, is not hindered much by this uninformative $\mathbf{z}$-space, as shown in Table 7. We attribute this to the effectiveness of preference modelling phase, where our method is still able to find a meaningful $\mathbf{z}^*$ in a less expressive $\mathbf{z}$-space.

This is also justified from t-SNE(Fig.6) as there our learned $\mathbf{z}^*$ (orange dot) locates just in the point of deep color.

| Environment | Dataset | OPPO(Ours) | OPPO(as) |
|---|---|---|---|
| | Medium-Expert | **107.8 ± 1.0** | 103.5 ± 4.4 |
| Hopper | Medium | **67.4 ± 3.5** | **69.2 ± 7.4** |
| | Medium-Replay | **88.2 ± 1.7** | 78.3 ± 7.1 |
| | Medium-Expert | **106.8 ± 2.8** | **108.8 ± 1.0** |
| Walker | Medium | **81.6 ± 2.8** | **80.7 ± 1.5** |
| | Medium-Replay | **70.6 ± 5.1** | 66.3 ± 1.6 |
| | Medium-Expert | **89.6 ± 0.8** | **90.1 ± 1.4** |
| HalfCheetah | Medium | **43.4 ± 0.2** | **43.4 ± 0.2** |
| | Medium-Replay | **39.8 ± 0.2** | **39.6 ± 0.1** |
| **Sum** | | **717.7** | 679.8 |

Table 7: Ablation study

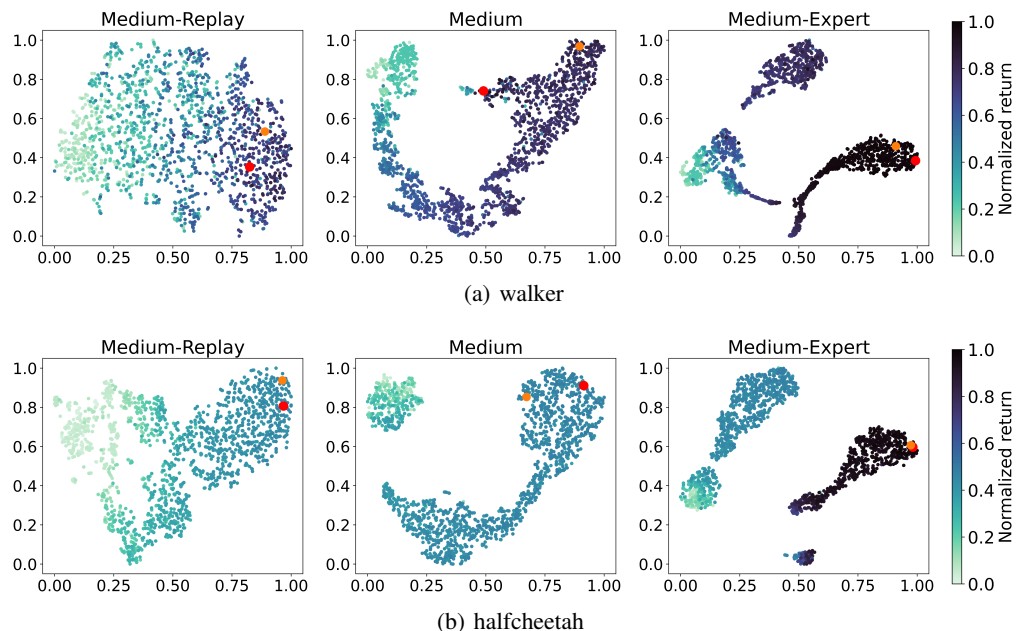

Figure 5: t-SNE of OPPO in walker and halfcheetah environments, including the embedding of random trajectories in $\mathcal{D}$, our learned $\mathbf{z}^*$ (orange dot) and the actual optimal $\mathbf{z}^{**}$ (red dot), embedding of the best trajectory/policy learned with online RL method. Color of the points represents the normalized return of the corresponding trajectory $\tau$.

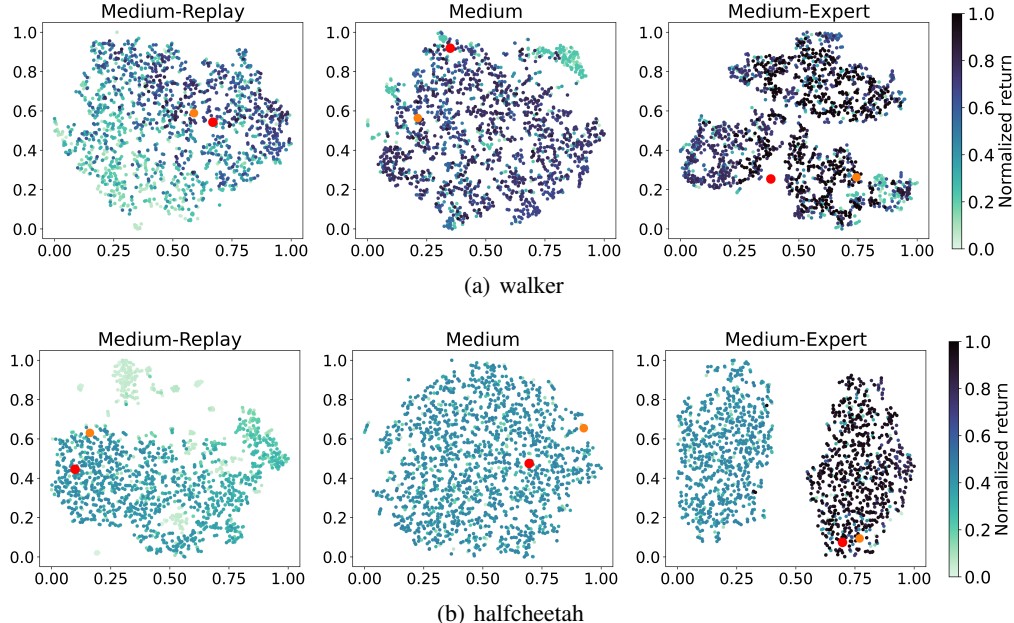

Figure 6: t-SNE visualization of the embedding space learned with OPPO-a in walker and halfcheetah environments.

**Ablation of feedback amount** For Hopper task, we evaluate the impact of different amounts of preference labels on the performance of OPPO as shown in Tab.8. More specifically, OPPO is evaluated using the labels amount from 50k, 3k, 1k, 500, on the dataset from Medium-Expert, Medium, Medium Replay. As illustrated in the table, OPPO performs the best when given 50k preference labels and achives a total normalized scoe of 283.1 among the three datasets. When

| Dataset | 50k | 3k | 1k | 500 |
|---|---|---|---|---|
| Medium-Expert | $108.0 \pm 5.1$ | $92.1 \pm 9.2$ | $102.9 \pm 3.2$ | $104.9 \pm 4.1$ |
| Medium | $86.3 \pm 3.2$ | $73.5 \pm 14.8$ | $90.8 \pm 2.0$ | $77.5 \pm 12.8$ |
| Medium-Replay | $88.9 \pm 2.3$ | $66.2 \pm 23.3$ | $60.4 \pm 3.0$ | $68.5 \pm 22.8$ |
| | 283.1 | 231.8 | 254.2 | 250.9 |

Table 8: Ablation study

the feedback amount decreases to 3k, the performance decreases at the same time. However, the performance stablizes at around 250 for further reduces to 1k and 500. Therefore, OPPO is robust to the changes in the amount of feedback used for training.

