# OpenReview forum: "Beyond Reward: Offline Preference-guided Policy Optimization"
_ICLR.cc/2023/Conference — Submitted to ICLR 2023_

### Official Review · Reviewer_LFVP · 2022-10-14

**Confidence:** 3
**Correctness:** 4
**Technical Novelty And Significance:** 3
**Empirical Novelty And Significance:** 4
**Recommendation:** 8

**Clarity, Quality, Novelty And Reproducibility:**

The approach is novel and relevant to the research area. The paper is of high quality and very good to follow, including the mathematical sections. However, to the reviewer it is not completely clear why $I_{\theta}$ is updated by (6) and (8) independently and not a joint objective. This way, it seems like, the magnitude of the updates may implicitly define a tradeoff between potentially conflicting targets?
BC in Tabl.2 is never introduced. Reproducibility is likely, as code is available.



**Strength And Weaknesses:**

The paper is very well written and clear to understand, with one, small exception (see clarity). The approach itself mainly builds on existing ideas, but goes beyond a simple, straight-forward combination. Furthermore, it is nicely aligned with the current State-of-the-Art and represents a meaningful step forward. The relation to related work is clearly state. The evaluation is ok, but could be improved. Mostly, because only one of the 4 competitors (Tbl.2) is a true PbRL algorithm. Furthermore, 3 domains (with 3 variants each) is ok, but leaves room for substantial improvement. Especially, non-locomotion tasks should be considered. Additionally, the required number of preference queries should be evaluated, as this is usually a limiting factor in PbRL (due to costly, human involvement). The qualitative ablation studies are a good addition to the evaluation.

**Summary Of The Paper:**

The authors introduce a novel approach for Preference-based Reinforcement Learning in an offline setting. In contrast to other methods, they do not learn an explicit reward function, but use information matching against a latent representation of an (approximate) optimal trajectory. The latent representation is learn via a transformer approach. The algorithm is evaluated on three benchmark domains, including analyzation of three specific aspects of the approach. Namely, an qualitative evaluation of the embedding space in terms of distance to an optimal trajectory, under a simplified learning scheme and effects of using different points in the latent space as conditioner.

**Summary Of The Review:**

The paper is very good and the clarity issue are not relevant for acceptance, but could improve the paper further. Only the evaluation is a relevant concern, but is still extensive enough to warrant a recommendation.

In fact, the reviewer believes, that the method is even more capable than mentioned. In theory, the method should be applicable to preference signals, based on a non-Markovian expert evaluations. This is an important issue for real world PbRL, because real humans cannot be assumed to conform to the Markov property. Therefore a discussion of this assumption (or even evaluation), would be quite interesting.

---

> ### Author Response · Authors · 2022-11-14
> **Response to Reviewer LFVP**
>
> Dear Review LFVP:
>
> Thank you very much for your time and effort. We would like to address your comment one by one in the following.
>
> **Q1: Regarding the baseline selections.**
>
> >**A1**: Please refer to the general response.
>
> **Q2: Regarding improving the evaluation and considering non-locomotion tasks.**
>
> >**A2**: We appreciate this valuable comment, and we are now gathering more experiment results. We will include them in the revised paper.
>
> **Q3: Evaluating the amount of preference queries.**
>
> >**A3**: Since improving the feedback efficiency is not the focus of this paper, we used a considerable amount of feedbacks(~50k labels) for all environments. We are conducting additional experiments to analyze the effect of different amounts of feedback on the performance of our method. However, we emphasize that unlike the online setting of previous works, large static datasets with preference labels can be easily obtained through crowd-sourcing platforms such as Amazon Mechanical Turk.
>
> **Q4: Why update $I_\theta$ independently by (6) and (8)?**
>
> >**A4**: As we described in the training objectives in Appendix 1, $I_\theta$ is updated by the weighted sum of the losses corresponding to the different objectives, and as you mentioned, we adjusted the weights as hyperparameters to tradeoff between the objectives.
>
> **Q5: Explanation for BC.**
>
> >**A5**: Sorry for the omission of its description, the following description will be added in the main body. BC is behavioral clone, which is a direct use of the dataset for supervised training of policy, and is a commonly used baseline in offline RL. The experimental results in the text we extracted from the original DT.
>
> **Q6: A discussion on the non-Markovian assumption**
>
> >**A6**: We believe that the non-Markovian hypothesis is a very worthwhile direction to explore and it is also more in line with the reality. We are also adding the corresponding experiments. However, we believe that the possible advantage of OPPO on non-Markovian tasks stems from the fact that our implementation uses transformer, a model commonly used for sequence modeling. We found that concurrent to our work, PT([https://openreview.net/forum?id=Peot1SFDX0](https://openreview.net/forum?id=Peot1SFDX0)) provides a more detailed discussion about this. Unlike their work, OPPO focuses more on the possibility of optimizing policies directly by preference rather than proxy rewards in the PbRL setting. In this framework, we can have more flexibility in the choice of models.

---

### Official Review · Reviewer_uVj7 · 2022-10-25

**Confidence:** 4
**Correctness:** 3
**Technical Novelty And Significance:** 2
**Empirical Novelty And Significance:** 2
**Recommendation:** 3

**Clarity, Quality, Novelty And Reproducibility:**

The idea is a creative combination of Hindsight Information Matching methods and preference-based reinforcement learning. It seems fairly novel, however it would have been nice to a discussion on the differences between the OPPO and Skill Preferences (https://proceedings.mlr.press/v164/wang22g/wang22g.pdf), both of which use preference feedback to learn latent embeddings used to condition a policy's behavior. In the case of Skill Preferences, the latent embeddings correspond to skills (sub-tasks) instead of complete tasks as in OPPO. Additionally in Skill Preferences, after the skill embeddings are learned, online PbRL is used to learn a policy to select a skill and used a preference-learned reward function. However, it would have been nice to see a discussion of the key differences in how the latent embeddings are learned.

From a reproducibility perspective, enough detail seems to have been provided that it should be possible to reproduce the results. However, I did not attempt to reproduce the results and therefore cannot say for certain. It would be great for the authors to release the code and make OPPO more reproducible. The hyper-parameter values for the number of preferences given does not appear to be given, but I may be missing it.

In terms of clarity and quality of writing, there are a number of places where the authors would benefit from another read through of the paper. There were many sentences with an awkward construction, e.g. "Sequence modeling enables to model behavior without....". While the construction did not make the paper incomprehensible, reducing the number of sub-clauses and parentheticals in each sentence would make the paper much easier to read and follow.

**Strength And Weaknesses:**

Strengths:
- The idea is really interesting. This is probably one of the first works I have seen that attempts to learn a policy from preferences in the absence of a reward function. Moving away from reward functions seems like it could have benefits and improve policy learning.
- The work is well grounded in the literature and authors do a good job of contextualizing where the method fits and how it is different from what has been done.
- A good amount of detail has been given to specify how OPPO works meaning the method and the results should be reasonable to reproduce.

Weakness:
- The proposed method does not consistently outperform the baselines. Clear benefits of the method are not well motivated, especially given the results. For example, the authors motivate their approach by stating that learning reward functions from preferences may result in a reward function that is not correctly calibrated to the preferences. The authors do not follow up with specific scenarios where or under which the reward functions are not well calibrated. It would have been nice to see clear cases where mis-calibration occurs when learning reward functions and then results demonstrating that OPPO is well calibrated in the identified scenarios.
- The authors only present results using a synthetic preference labeller that gives perfect preference feedback with respect to the ground truth reward. However, humans are not likely to provide such high quality labels. As the quality of the offline dataset impacts rollout performance, it is likely that preference label quality will also impact rollout performance. It has been shown in the BPref (Lee et al. 2021) paper that labeller quality does impact policy performance.
- I did not see a discussion about the impact of the amount of preference feedback on rollout performance. In online PbRL methods where the reward function is learned, the amount of preference feedback greatly impacts policy performance. It would have been great to see results on this and for there to have been a discussion.
- The authors claim that their method is guaranteed to learn the optimal latent embedding z, however no proof that optimality is guaranteed is provided. The experiments (i.e. t-SNE plots) are not enough evidence to conclude optimality is guaranteed.
- Prior online PbRL papers have included results from human experiments where actual humans provide the preference feedback. It would have been nice to see results with humans providing the preference feedback.
- The authors do not appear to have compared against Shin & Brown 2021, who they call out as a very similar method that learns a reward function from preferences over an offline dataset and then use offline RL methods to learn a policy. It may have been present in the baselines, but it wasn't clear to me.

**Summary Of The Paper:**

The paper presents an approach for learning to solve task from offline data using preference feedback and no reward function. Instead of learning a reward function, a latent embedding is learned that encode information about what it means to complete the target task. The latent embedding, z, is learned from preferences over the offline data used to train the policy. The latent embedding z and the policy are learned separately, where there during policy learning and inference, next actions are conditioned on both z and the current observation. Policy training is conducted following Hindsight Information Matching. Experiments are conducted in the Hopper, Walker, and Half cheetah locomotion tasks from D4RL with offline data coming from the replay buffer of a medium quality policy, rollouts from a medium quality policy, and rollouts from a medium-expert quality policy. To evaluate the quality of the learned latent embedding z*, a t-SNE plot is used to visualize how similar z* is a an ideal trajectory. Across conditions, z* is close to the ideal trajectory. The performance of policy rollouts using the "optimal" z embedding, a z embedding from a trajectory with large returns, and a z embedding with low returns are compared. Across conditions rollouts from the "optimal" z embedding out perform the z embedding from the trajectories with high and low returns. Policy rollout performance is also compared against other offline RL approaches (decision transformer with the ground-truth reward, decision transformer with a learned reward, CQL with the ground-truth reward, and behavior cloning). In 1/3 of the conditions, the proposed methods out perform the baselines. Finally, the impact of iterating between the Hindsight Information Matching objective and policy learning is evaluated via t-SNE projections and visualization. In all but the medium condition, the learned z* and the ideal trajectory are further apart than in the first experiment.

**Summary Of The Review:**

In summary, the paper presents an interesting idea that seems of benefit to pursue. However, given that OPPO did not consistently outperform the baselines, better motivation for why the method should be used instead needs to be provided. The benefits of OPPO over the baseline need to made more clear. Additionally, there are further experiments that would be helpful to see to fully understand the benefits and robustness of the method. My recommendation for the paper is therefore based upon the need for more experiments and the lack of consistent improvements in performance relative to baselines.

---

> ### Author Response · Authors · 2022-11-15
> **Response to Reviewer uVj7 (2/2)**
>
> **Q6: Comparing against Shin & Brown 2021.**
>
> >**A6**: The main contribution of OPAL ([https://arxiv.org/pdf/2107.09251.pdf](https://arxiv.org/pdf/2107.09251.pdf)) is being the first to introduce the offline setting to PbRL, while OPAL used the traditional two-step approach and did not propose a novel solution. Further, OPAL reported that relabeling offline trajectories via “a trivial reward function” may not lead to a policy degradation, and concluded that “many offline RL benchmarks are ill-suited for reward learning”. Such phenomenon are also confirmed in UDS ([https://arxiv.org/pdf/2202.01741.pdf](https://arxiv.org/pdf/2202.01741.pdf)). Our intuition aligns with this in the sense that OPPO is optimizing the policy directly via preferences instead of an **intermediate reward learning phase**. We choose to use DT+$r_\psi$ to represent for the traditional two-stage baselines including OPAL for two reasons. Firstly, OPAL has not released their source code yet. And secondly, the evaluations only included the environments that they believe have necessity to do reward learning.
>
> **Q7: A discussion on the differences between OPPO and SkiP**
>
> >**A7**: SkiP is hierarchical RL framework aiming to learn a set of pretrained skills in the offline setting, while the high-level $z$-space for the downstream task is learned in the online setting. In contrast, OPPO is a fully offline solution. During the pretraining stage of SkiP, the $z$-space is learned via labels “whether a trajectory is noisy or structured”, which are different from the labels for downstream tasks, and thus task-irrelevant. On the other hand, the learning of our $z$-space incorporates the task preferences, and hence the learning of the embedding space and the optimal embedding is interdependent.
>
> **Q8: "It would be great for the authors to release the code and make OPPO more reproducible."**
>
> >**A8**: We submitted the zip file of our source code in the supplemental material at the time of the main submission and highlighted this in the first paragraph of the appendix. Besides, the other reviewers acknowledged the reproducibility of our work. We strongly recommend the review to have a check on our code.
>
> **Q9: Regarding the writing quality**
>
> >**A9**: We apologize for any confusion caused by the writing, and we are in the process of proofreading. The paper quality will be improved from the perspective of grammar and sentence constructions, we promise.

---

> ### Author Response · Authors · 2022-11-15
> **Response to Reviewer uVj7 (1/2)**
>
> Dear Review uVj7:
>
> Thank you very much for your time and effort. We would like to address your comment one by one in the following.
>
> **Q1: Regarding the performance and benefits of OPPO.**
>
> >**A1**: Please refer to the general response.
>
> **Q2: Why don't training with actual human feedback, since label qualities would impact rollout performance?**
>
> >**A2**: We agree that human-labeled preferences may include noise, incorrectness and non-Markovian.  The performance of strategy is certainly up to the labelling quality, and it is a universally concerned problem for the community. But the main focus of our paper is on the issue whether we should learn a reward function explicitly or not. The quality of the preference label brings a performance degradation that we believe is fair for both. Given that it is already widely accepted in PbRL work to use oracle scripted teachers for evaluation, such as PEBBLE([http://proceedings.mlr.press/v139/lee21i.html](http://proceedings.mlr.press/v139/lee21i.html)), SURF([https://openreview.net/forum?id=TfhfZLQ2EJO](https://openreview.net/forum?id=TfhfZLQ2EJO)), and RUNE([https://openreview.net/forum?id=OWZVD-l-ZrC](https://openreview.net/forum?id=OWZVD-l-ZrC)), we believe that it is acceptable that we do not provide an analysis of this issue in this paper. Of course, while this may be out of our scope, if a benchmark based on real human preferences is released in the future, we would be happy to evaluate OPPO on it.
>
> **Q3: Regarding the impact of the amount of preference queries.**
>
> >**A3**: Since improving the feedback efficiency is not the focus of this paper, we used a considerable amount of feedbacks(~50k labels) for all environments. We are conducting additional experiments to analyze the effect of different amounts of feedback on the performance of our method. However, we emphasize that unlike the online setting of previous works, large static datasets with preference labels can be easily obtained through crowd-sourcing platforms such as Amazon Mechanical Turk.
>
> **Q4: A guarantee for optimality.**
>
> >**A4**: As shown in Figure 3 (Section 5.1), the embedding learned by OPPO is located in the cluster with high rewards. And it is close to embedding of the optimal trajectory in offline dataset. As shown in Table 1 (Section 5.2), the trajectory learned by OPPO constantly outperforms the trajectory stored in the offline dataset. Therefore, such results empirically confirm that the embedding learned by OPPO is optimal. In the setting of PbRL, it is an unsolved problem to define optimality, as the supervision provided via preferences are relative not absolute. The maximum of “cumulative proxy rewards” is only a proxy objective for tasks defined by preferences. In previous papers using two-stage solutions[1-6], none of them have theoretical guarantees that the policy maximizing the relabeled reward is equivalent to the one most desired by the human labeler. What they have instead is carefully designed and conducted experiments to show that their approaches outperformed the baselines. Hence, it is both hard and unfair to require a theoretical guarantee for OPPO.
>
> [1] Christiano, Paul F., et al. "Deep reinforcement learning from human preferences." *Advances in neural information processing systems* 30 (2017).
>
> [2] Ibarz, Borja, et al. "Reward learning from human preferences and demonstrations in atari." *Advances in neural information processing systems* 31 (2018).
>
> [3] Brown, Daniel, et al. "Extrapolating beyond suboptimal demonstrations via inverse reinforcement learning from observations." *International conference on machine learning*. PMLR, 2019.
>
> [4] Lee, Kimin, Laura Smith, and Pieter Abbeel. "Pebble: Feedback-efficient interactive reinforcement learning via relabeling experience and unsupervised pre-training." *arXiv preprint arXiv:2106.05091* (2021).
>
> [5] Park, Jongjin, et al. "SURF: Semi-supervised Reward Learning with Data Augmentation for Feedback-efficient Preference-based Reinforcement Learning." *arXiv preprint arXiv:2203.10050* (2022).
>
> [6] Liang, Xinran, et al. "Reward Uncertainty for Exploration in Preference-based Reinforcement Learning." *arXiv preprint arXiv:2205.12401* (2022).

---

### Official Review · Reviewer_JqyW · 2022-10-25

**Confidence:** 4
**Correctness:** 2
**Technical Novelty And Significance:** 2
**Empirical Novelty And Significance:** 2
**Recommendation:** 3

**Clarity, Quality, Novelty And Reproducibility:**

The algorithm is new. Code was submitted, but I did not run it. The writing needs to be improved. There are too many informal arguments.

**Strength And Weaknesses:**

Strength:
- The problem setup in this paper is more realistic than most of the offline RL papers, where the reward is always available for each state-action pair.

- The proposed algorithm is new and can greatly improve the performance over baseline methods on the D4RL benchmark.

Weakness:
- The motivation of OPPO is unclear to me. Especially, why optimizing two (heuristic) objectives iteratively is a better choice than solving two separated optimizations, i.e., first, learn a reward function based on the preference, and then fit a policy. Can you provide a more rigorous justification for it?

- In addition, I also noticed that DT+$r_{\psi}$ performs quite close to (only slightly worse than) OPPO, but it performs slightly better than DT+r (ground-truth reward). This is quite surprising, can you explain why this is the case? Also, it seems the OPPO has little advantage over the two-stage algorithm, DT+$r_{\psi}$.

- The paper is not easy to follow, it has too many vague, informal and unsupported arguments. E.g.,

> "OPPO learns the optimal (offline) policy (π(a|s, z∗)) directly and thus avoids the potential information bottleneck caused by separately learning a reward function"  what do you mean by information bottleneck?

> "A better estimate of the optimal embedding provides hints for the encoder to extract more valuable features, while a better hindsight information encoder, on the other hand, accelerates the search process for the optimal trajectory in the high-level embedding space":  what do you mean by "hints"? Can you provide evidence for supporting this argument?

> "It is worth mentioning that the posterior of the optimal embedding $z^\star$: why call it a posterior? Shouldn't $z^\star$ just be a point?

- What is $\ell$? I did not see the definition of it.

- Is $\pi(\cdot|s,z)$ a parameteric model?

- In general, offline RL relies on pessimism to overcome the distribution shift. It seems that OPPO doesn't need pessimism to perform well. If so, can you explain why it doesn't need?

- Can you provide a comparison on the *Random* datasets, e.g., Hopper-Random, Walker-Random, and Halfcheetah-Random? Also, why is the Ant dataset missing?

**Summary Of The Paper:**

This paper studies preference-based offline reinforcement learning where we only have access to the preferences over offline trajectories. Compared to the standard offline RL setting, PbRL doesn't assume the reward function is available, which is more realistic. To solve the problem, the authors propose an iterative algorithm, which optimizes two objectives iteratively and can directly output a policy, instead of first learning a reward function from the preferences and then training a policy on top of it. Empirically, the authors compare their algorithm with several baselines on the D4RL benchmark and demonstrate improved performance.

**Summary Of The Review:**

Overall, I think the problem studied in this paper is very interesting and close to reality. However, I am not really convinced with the motivation of the proposed algorithm, and why we need to do so as in this paper. In addition, the writing of this paper needs to be improved. There are too many vague/ambiguous and informal arguments. The authors should make explainations more formally, and provide either theoretical justifications or empirical results to backup their arguments.

---

> ### Author Response · Authors · 2022-11-14
> **Response to Reviewer JqyW**
>
> Dear Review JqyW:
>
> Thank you very much for your time and effort. We would like to address your comment one by one in the following.
>
> **Q1: Why is optimizing iteratively better than separately solving them?**
>
> >**A1**: Please refer to the general response.
>
> **Q2: Why DT+$r_\psi$ performs close to OPPO but slightly better than DT+$r$? What is our advantage over the two-stage algorithm?**
>
> >**A2**: The DT+$r$ experimental results in our table are from the DT paper ([https://papers.nips.cc/paper/2021/hash/7f489f642a0ddb10272b5c31057f0663-Abstract.html](https://papers.nips.cc/paper/2021/hash/7f489f642a0ddb10272b5c31057f0663-Abstract.html)). The performance gap may stem from some implementation details, regarding which we are reproducing for verification.
>
> **Q3: Vague arguments:**
>
> 1. What does it mean by the information bottleneck?
>
> >**A3.1**: Please refer to the general response.
>
> 2. What does it mean by the hints?
>
> >**A3.2**: When $z^*$ learned by OPPO approaches the most preferred embedding by the human labeler, accordingly updating the $z$-space restores the underlying $z$-space modeled by the human labeler. You can refer to the ablation studies that when the information extractor is not optimized using the preference modeling loss calculated based on the updated $z^*$, the performance of OPPO degrades.
>
> 3. Shouldn’t $z^*$ be a point?
>
> >**A3.3**: Yes. We optimize $z^*$ as a point in our experiment, but we could also model it as a point sampled from a distribution, and optimize the distribution.
>
> **Q4: The definition of $\ell$**
>
> >**A4**: $\ell$ is the loss function which is mentioned in the last sentence of the first paragraph in Section 3.2. and we specifically describes the setup during the experiment in the paragraph of training objective in Appendix A.1. Sorry that there is no further clarification in the main body. We consciously vague this definition to reserve flexibility for $\ell$ based on different definitions of $z$. In our actual implementation, $z$ is a point in the contextual space, and thus $\ell$ is the Euclidian distance. On the other hand, $z$ could also be a point sampled from a learned distribution, and hence in such a case $\ell$ is a measurement between two distributions such as KL divergence. In addition, we perform certain symbolic abuse on Eq. (5) and (6). Specifically, in $\ell(\tau,\tau_z)$, $\ell$ denotes the accumulation of MSEloss on the trajectory between the $a$ obtained using $\pi$ on each $s$ in the trajectory and the true $a$ in the data set, i.e., the reconstruction error on the trajectory. Thank you for the comment, and the corresponding revision will be reflected.
>
> **Q5: Is $\pi$ a parametric model?**
>
> >**A5**: Yes. As mentioned in the last sentence of the first paragraph in Section 4.3, the $z$-conditioned policy is implemented using GPT, where the input is $s$, $z$, and the output is $a$. During training, the parameter of GPT is updated via equation (6), while $z$ is the embedding extracted by $I_\theta$  based on the dataset. During test time, the parameters of GPT is fixed and the goal is to find the optimal embedding $z^*$.
>
> **Q6: How could OPPO deal with the OOD issue?**
>
> >**A6**: Currently, there are two major solutions to overcome the distributional shift in the actor-critic architecture. The first is to introduce pessimism to learn a conservative estimation for OOD state-action pairs. The second is to regularize the actor so that the policy stay close to the behavior policy. As OPPO is an extension of HIM, our method bears a resemblance to the latter. Consequently, the reconstruction loss ensures the distribution of the policy is consistent to the distribution of the offline trajectories.
>
> **Q7: Regarding the Random datasets and the Ant dataset.**
>
> >**A7**: We appreciate this valuable comment, and we are now gathering more experiment results. We will include them in the revised paper. As DT was not evaluated on the Random and Ant Dataset, we did not include them in the first place.

---

### Official Review · Reviewer_kZ4s · 2022-10-25

**Confidence:** 2
**Correctness:** 3
**Technical Novelty And Significance:** 3
**Empirical Novelty And Significance:** 2
**Recommendation:** 6

**Clarity, Quality, Novelty And Reproducibility:**

The method seems like a clear extension of HIM, but seems novel enough. The algorithm 1 is confusing. It seems reproducible enough.

**Strength And Weaknesses:**


Generally I like this research direction, and I think that learning context from offline data can be useful for other tasks as well, such as learning options
 Unfortunately, it doesn’t look like the method convincingly outperforms existing baselines. However, it does perform competitively, and I think that the fact that it doesn’t learn a reward model is a nice aspect.
I like the experiments showing that the learned optimal context aligns with preferences.
A question: what is the main difference between learning a reward model and learning a context + the optimal context? It seems almost the same thing. Could they be thought of as the same thing in a broader unified perspective?  Why would we expect OPPO to perform better than learning a reward model?



**Summary Of The Paper:**

This paper looks at the problem of offline RL from human preferences. Instead of first learning a reward model from human preferences and then training with offline RL, this paper learns a context embedding from trajectories, and then learns a policy that is conditioned on the context. Then, to optimize for the human preferences, it learns the optimal context, which can be fed into the policy.


**Summary Of The Review:**

I think this is a nice idea, and it should be useful for the community.

---

> ### Author Response · Authors · 2022-11-14
> **Response to Reviewer kZ4s**
>
> Dear Review kZ4s:
>
> Thank you very much for your time and effort. We would like to address your comment one by one in the following.
>
> **Q1: The difference between the reward-learning PbRL and ours. Why expect a better performance?**
>
> >**A1**: Please refer to the general response.
>
> **Q2: Regarding algorithm 1.**
>
> >**A2**: Based on an offline dataset of trajectories, OPPO returns a latent-conditioned policy and the optimal contextual embedding. In Line 4, we first use the hindsight information matching loss (equation 6) to train the policy network and the hindsight information extractor. Consequently, given the $z$ extracted out of an offline trajectory by the extractor, the policy is able to reconstruct it. In Line 6, using trajectory pairs and the corresponding preference labels $(z^+, z^-, y)$ , we iteratively update the optimal embedding and the information extractor via the preference modelling loss (equation 8). The objective is to make the optimal embedding $z^*$ to be near to the more preferred trajectory $z^+$, and meanwhile further away from the less preferred trajectory $z^-$. And more clarification upon the method will be included in the revision.

---

### Author Response · Authors · 2022-11-14
**General response to all Reviewers**

Dear all reviewers,

We are grateful for your time and elaborate comments which we find very helpful and constructive. Meanwhile, we are honored to have the acknowledgment of novelty from all of you. In this general response, we would like to highlight the primary motivation of our work.

**In preference-based reinforcement learning (PbRL) setting,  tasks are defined by preference labels.**

The previous works in PbRL use a two-step method, which firstly trains a scalar reward function via supervised learning from preference label, and then uses reinforcement learning algorithms to train a policy based on the relabeled trajectories. The policy optimization aims to maximize the cumulative discounted proxy rewards of the policy rollouts. We argue that conveying information from preferences to the policy via the scalar rewards is the bottleneck in such process. On one hand, when encountering complex tasks (such as non-Markovian information that preference may contain), the information capacity of scalar reward assignment may be limited. On the other hand, such a two-step procedure is subject to noisy reward labels. More specifically, when the learned reward function is imperfect, noise would be introduced to policy learning.

Instead of training a scalar reward function, we propose a new paradigm of learning policies in a high-dimensional $z$-space. We first learn the $z$-space capturing more task-related information, and then search for the optimal $z$ to determine the policy conditioned on $z$. Our intuition is similar to introducing successor features into PbRL like Regret Preference did ([https://arxiv.org/pdf/2206.02231.pdf](https://arxiv.org/pdf/2206.02231.pdf)), while our focus is on learning $z$-space and evaluating policies in such a high-dimensional space. We hope that OPPO can draw the community's attention to the potential of this new paradigm, especially for its applications on reward-free tasks (like PbRL). However, due to the time and computational resources limitations, the selection of models and search methods requires more future research.

---

### Author Response · Authors · 2022-12-07
**Summary of Revision**

We would like to thank the reviewers for their detailed and insightful comments again.
In this paper, we employ hindsight information matching (HIM), a supervised learning paradigm, on offline PbRL problems.
We use contrast learning to model the preference for both searching the optimal information ($z^*$) used in the evaluation phase and constraining the training of HIM.
The new paradigm dispenses with learning a separate reward function and directly optimizes the policy using preferences.

We have revised our paper to accommodate the concerns raised by the reviewers. The paper modifications are highlighted in red and we provide a list in the following:
- We strengthened our motivation for direct policy optimization in the introduction as discussed above.
- We clarified the design consideration regarding $z^*$ and its connection to the most preferred offline trajectory in Section 4.
- We replaced the ambiguous statement regarding “information bottleneck” and “features extracted by the encoder” in Section 4.2.
- We provided more details for the pseudo code, especially for Line 4-6.
- We conducted more experiments to empirically show the competitive performance of OPPO in mujoco tasks. The results are updated in the table accordingly in Section 5.3.
- We moved the implementation details (framework, architecture,  design choice of context) to Appendix.
- We ablated the amount of feedback, and the results are appended in Appendix. The results shows that the performance of OPPO is robust to the decrease of human preference labels.

---

### Decision · Program_Chairs · 2023-01-20

**Decision:**

Reject

**Justification For Why Not Higher Score:**

The reviewers rating are 6 / 3 / 3  / 8. The review of the reviewer giving a score of 6 is relatively superficial and the confidence of the reviewer is low, so I discounted this review. The remaining ratings show a relatively clear signal for rejection and also the most positive reviewer agreed in the discussion to the concerns of the negative reviewers.

**Justification For Why Not Lower Score:**

n/a

**Metareview: Summary, Strengths And Weaknesses:**

The paper consider an offline preference-based reinforcement learning setting and propose a method for learning a behavioural policy in an end-to-end fashion avoiding a naive two-stage strategy in which one first infers a reward function and then performs a standard approach for offline reinforcement learning. The authors claim that this is beneficial "when encountering complex tasks" (e.g., with non-Markovian information in the preferences) and when there are noisy preference ("a two-step procedure is subject to noisy reward labels") because in such a case "the learned reward function is imperfect, [and] noise would be introduced to policy learning".

The reviewers agreed that these claims are sensible and an end-to-end approach for policy learning in general useful. However, the authors fail to show that these claims are indeed true. In particular, no experiments are presented which demonstrate the claimed advantages of their approach. The authors should present experiments with non-Markovian rewards and noise in the preference feedback to show that their approach really has the claimed advantages. Additionally understanding how the method behaves in terms of the amount of preference feedback provided would be important to study. Furthermore, the writing of the paper should be improved. Not all details of the proposed approach are easy to comprehend from the given writeup, e.g., the possible meaning of the losses, information bottleneck, etc. Some of these problems can be fixed by moving material from the appendix to the main paper, some others might require a more careful rephrasing of the authors' approach.

In summary: while the authors' approach is promising and sensible, the current evaluation of the approach is not convincing and should be improved for future submissions. Furthermore, the paper and proposed method would be more accessible if clarity of the paper was improved.

**Summary Of Ac-Reviewer Meeting:**

We had a virtual discussion. Reviewers LFVP and JqyW were present, reviewers uVj7 and kZ4s could not participate.

We discussed the reviewers’ view on the paper and their considerations of the authors’ responses. While the authors’ responses clarified several details about the paper for the reviewers, it did not change their general perception of the paper. While the reviewers recognise the paper’s setting and approach as sensible and promising, the authors fail to demonstrate the advantages of their approach against two-step based preference-based RL. In particular, the authors should demonstrate the advantage of their model by considering problems with non-Markovian rewards and noise in the preference feedback (points also brought up by the authors in the rebuttal as potential strengths of their method). Additionally understanding how the method behaves in terms of the amount of preference feedback provided would be important to study.

In summary, we concluded that given the current reviews, rebuttal, and discussion, the paper currently does not meet the bar for acceptance. Of course further discussion until the deadline on Monday can still change this.